# In Situ Analysis of Interactions between Fibroblast and Tumor Cells for Drug Assays with Microfluidic Non-Contact Co-Culture

**DOI:** 10.3390/mi9120665

**Published:** 2018-12-17

**Authors:** Hongmei Chen, Wenting Liu, Bin Wang, Zhifeng Zhang

**Affiliations:** 1School of Mathematics and Physics of Science and Engineering, Anhui University of Technology, Maanshan 243002, China; 2Division of Nanobionic Research, Suzhou Institute of Nano-Tech and Nano-Bionics, Chinese Academy of Sciences (CAS), Suzhou 215123, China; wentingliu@sinano.ac.cn; 3University of the Chinese Academy of Sciences, Beijing 100049, China; 4Department of Engineering Science and Mechanics, The Pennsylvania State University, State College, PA 16802, USA

**Keywords:** laminar flows, paracrine signaling, co-culture

## Abstract

Fibroblasts have significant involvement in cancer progression and are an important therapeutic target for cancer. Here, we present a microfluidic non-contact co-culture device to analyze interactions between tumor cells and fibroblasts. Further, we investigate myofibroblast behaviors induced by lung tumor cells as responses to gallic acid and baicalein. Human lung fibroblast (HLF) and lung cancer cell line (A549) cells were introduced into neighboring, separated regions by well-controlled laminar flows. The phenotypic behavior and secretion activity of the tumor cells indicate that fibroblasts could become activated through paracrine signaling to create a supportive microenvironment for cancer cells when HLF is co-cultured with A549. Furthermore, both gallic acid (GA) and baicalein (BAE) could inhibit the activation of fibroblasts. In situ analysis of various cell communications via the paracrine pathway could be realizable in this contactless co-culture single device. This device facilitates a better understanding of interactions between heterotypic cells, thus exploring the mechanism of cancer, and performs anti-invasion drug assays in a relatively complex microenvironment.

## 1. Introduction

Cancer progression has an affinity relationship with the tumor microenvironment, including the extracellular matrix (ECM), fibroblasts, immune cells, and endothelial cells, and the blood vessels and proteins produced [1,2]. Fibroblasts synthesize components of the extracellular matrix (ECM), thus forming the structural framework of the tumor microenvironment. Fibroblasts are normally quiescent. Under some conditions, fibroblasts could become activated and acquire an activated phenotype. Activated fibroblasts located adjacent to cancer cells as carcinoma-associated fibroblasts (CAFs) could support tumor epithelial growth and invasion [3]. On one side, normal fibroblasts have mostly been thought to play a more passive role in cancer. On the other side, some researchers compared the effect of normal fibroblasts with that of CAFs on tumor cells and demonstrated that the former inhibit cancer progression [4,5]. Other researchers found that normal fibroblasts could induce tumor growth [6,7,8]; further, quiescent fibroblasts can become activated and might be key regulators of paracrine signaling during cancer progression [9]. Therefore, it is necessary to understand the mechanism between normal fibroblasts and tumors for personalized, targeted anticancer therapies.

The traditional co-culture platforms for studying cell–cell communications are limited to culture dishes and Transwell assays, which cannot avoid the drawbacks of complicated manual operations and large consumption of reagents. Microfluidic chips can integrate various experimental operations and mimic an in vivo microenvironment [10]. Over the past decade, considerable progress has been made in culturing cells with microfluidic chips [11,12]. Xie et al. reported a microchip that achieved co-culture and made a “wound” by removing a narrow barrier after cell seeding [13]. This device could co-culture different types of cells for cell migration assay. However, physical removal of a cell monolayer could damage cells, thus affecting the experimental result of cell migration. Businaro et al. realized an on-chip model consisting of two center end-closed channels to investigate the interactions between cancer and the immune system [14]. This model elucidated that the reciprocal interactions were heterogeneous; however, the model was difficult to process. Besides this, typical cell–cell communication can be classified into direct and indirect contact modes [15]. To date, much research has been performed on the direct mode, but focus on the indirect mode has been rare. It is not clear whether cells affect each other through cell–cell contacts or paracrine signals in the direct mode, but cells could only communicate through diffusible signals secreted in the indirect.

To overcome these limitations, we developed a microfluidic device with a novel technique, well-controlled laminar flow, allowing two types of cells to be in non-contact and only experience paracrine interactions with limited amounts of reagent consumption. A similar design has been presented previously [16,17]. Cellular responses can be observed immediately without physical damage and distinguished after fluorescent labeling. With this device, we are able to investigate interactions between tumor cells and normal fibroblasts and analyze the function of antitumor drugs with different concentrations in tumor-induced fibroblasts. This rarely reported co-culture mode provides a cost-effective approach to accomplish multiple functions, including cell loading with passive pumping, heterogeneous cell compartmentalization, and an accurate and reliable cellular assay directly monitored in real time. The precise and well-controlled laminar flow makes it a perfect non-contact co-culture platform. Moreover, multiple types of cells can be contactlessly co-cultured by designing several entrance channels, which enables us to mimic complicated microenvironments in vivo. Thus, it provides a compartmentalized co-culture model to elucidate reciprocal interactions between heterogeneous cell types within tumors, posing a relevant impact on antitumor therapeutic strategies.

## 2. Materials and Methods

### 2.1. Design and Fabrication of the Microfluidic Co-Culture Device

The microfluidic device was composed of a polydimethylsiloxane (PDMS, Silgard 184, Dow Corning, Midland, MI, USA) layer and a glass substrate. The upper layer, in Figure 1a, which contained three inlet channels (5 mm × 300 μm × 100 μm) that converged into a main channel (5 mm × 900 μm × 100 μm), was fabricated using PDMS following a well-established replica molding process. A mold with micrometer-sized structures was prepared by standard soft lithography methods. A mixture of PDMS followed the manufacturer’s instruction was poured onto the mold at 85 °C for 45 min. After cooling, the cured PDMS layer with the desired structures was gently peeled off from the mold and punched to form inlets and an outlet. Then, PDMS was bonded to the glass slide after oxygen plasma treatment for 50 s. 

Before use, the device was sterilized with 70% (*v*/*v*) ethanol for 1 h and exposed to UV light for 30 min. 

### 2.2. Formation of Laminar Flow in Microchannels

As shown in Figure 1b, samples containing different kinds of cells were placed into the three inlets. Continuous flow was obtained due to gravitational forces created by pressure differences from a fluid level discrepancy between the inlets and outlet. In order to confirm laminar flow formation inside the main microchannel, rhodamine B (RB), PBS solution, and blue ink (BI) were injected into the different inlets. A paper tip was placed at the outlet to remove the waste and continuous laminar flow of all solutions was maintained toward the outlet. The laminar flow was observed using an inverted microscope (Nikon Ti-Eclipse, Tokyo, Japan).

### 2.3. Cell Culture

A549 cells were purchased from ATCC, and human lung fibroblast (HLF) cells were kindly offered by Dr. Yimin Zhu (Suzhou Institute of Nano-Tech and Nano-Bionics, CAS, Beijing, China). Cells were cultured in RPMI-1640 (Hyclone Corp., San Angelo, TX, USA) supplemented with 10% (*v*/*v*) fetal bovine serum (FBS; Hyclone Corp., San Angelo, TX, USA), 100 U/mL penicillin, and 100 U/mL streptomycin. A culture incubator with a humidified atmosphere of 5% CO_2_ and at 37 °C was used for cell culture. 

### 2.4. Co-Culture of A549 and HLF on the Microfluidic Device

Cells were harvested using 0.25% trypsin/EDTA (Hyclone Corp., San Angelo, TX, USA) when they reached 70%–80% confluence and were gently re-suspended in cell culture medium at a density of 5 × 10^6^ cells/mL. In order to investigate interactions between tumor cells and normal fibroblasts, cells were divided into experimental and control groups. In the experimental group, 3 μL HLF and 3 μL A549 were separately and simultaneously injected into inlets on both outer sides. After a while, when both cell types had settled down, 6 μl of cell culture medium without cells was loaded from the middle. HLF and A549 went straight along the channel, without any mixture, and finally formed a non-contact co-culture model. In the control groups, homogeneous cell models of HLF–HLF and A549–A549 were cultured in the microchannels. Cells were grown on the surface of the glass substrate inside the microchannels, and cell morphologies were monitored using an inverted microscope (Nikon Ti-Eclipse, Tokyo, Japan).

### 2.5. Cell Viability Assay

Cells with a density of 5 × 10^6^ cells/mL were loaded into the device and driven into the disjunctive side areas of the main microfluidic channel. Then, the device was placed into a cell culture incubator. After 48 h, a cell viability assay was performed with two specific fluorescent probes: Hoechst 33342 (Molecular Probes, Solarbio Corp., Beijing, China) and propidium iodide (PI, Molecular Probes, Solarbio Corp., Beijing, China). Briefly, we introduced 5 μg/mL of each reagent into the main channel and incubated the device at 4 °C for 20 min. Images were acquired using an inverted fluorescent microscope (Nikon Ti-Eclipse, Tokyo, Japan). Cell viability was statistically quantified by the proportion of living cells to total cells from several optional fields.

### 2.6. Cell Migration on the Microfluidic Co-Culture Device

In order to investigate how fibroblasts and tumor cells interact, the boundary perimeter of the regions containing migrated cells was captured to record cell positions using an inverted microscope (Olympus CKX41-A32PH, Tokyo, Japan) every 4 hours after HLF and A549 were compartmentalized into opposite sides of the main passage. The migration distance is defined as average length of all cells migrating into the blank space between the two compartments and can represent the migration ability of cells. Migration distances can be quantified at different times. 

### 2.7. Immunocytochemistry and Fluorescence Imaging

To identify whether HLF was activated, expression of α-SMA in HLF was recognized by an immunofluorescence technique. After 2 days of co-culture, cells were fixed in 4% paraformaldehyde for 20 min, rinsed twice in PBS, and permeabilized in 0.25% Triton X-100 for 15 min. After three washes in PBS, nonspecific hybridization was blocked in 5% Bovine Serum Albumin (BSA) for 40 min at 37 °C. Cells were immunostained with primary antibody (mouse anti-α-SMA, Boster Corp., Wuhan, China) at 1:200 overnight. Then, they were incubated with secondary antibody (FITC-labeled goat antimouse IgG, Boster Corp., Wuhan, China) at 1:46 dilution for 50 min. After washing in PBS, they were incubated with 5 μg/mL Hoechst 33342 staining nucleus for 20 min. Images were taken using an inverted fluorescent microscope (Nikon Ti-Eclipse, Tokyo, Japan).

### 2.8. Enzyme-Linked Immunosorbent Assay

Cells were cultured in the chip, and then the supernatant was collected at the outlet after 24 h and centrifugated at 5000 rpm for 10 min. To measure the level of TGF-β1 released in the medium, an enzyme-linked immunosorbent assay (ELISA) was performed using commercially available kits for cytokine detection (Human TGF-β1 ELISA kit Boster Corp., Wuhan, China). The optical density was determined within 30 min at 450 nm on an ELISA microplate reader (Multilabel Plate reader PerkinElmer, Waltham, MA, USA).

### 2.9. Anticancer Drug Assays on the Microfluidic Co-Culture Device

It was elucidated that gallic acid (GA) or baicalein (BAE) could interfere with the interactions between HLF and A549 cells. After cells were co-cultured for 8 h, drug media with different concentrations of GA or BAE varying from 0 to 80 μg/mL were introduced into the co-culture device. After 24 h of incubation, apoptosis of HLF cells was analyzed after treatment with GA or BAE. The migration ability and expression of α-SMA in HLF cells were analyzed.

### 2.10. Statistical Analysis

All these experiments were replicated three or more independent times, and data are presented as the mean ± standard deviation (SD). Distances of fibroblast migration and comparisons with controls were evaluated statistically using Student’s *t*-test. A value *p* < 0.05 was considered statistically significant.

## 3. Results and Discussion

### 3.1. HLF and A549 Indirect Co-Culture on the Microfluidic Device

In this work, a non-contact co-culture microfluidic platform was developed for the study of interactions between tumor cells and fibroblasts (Figure 2). Three branches converging to a single main microchannel were subjected to laminar fluid flow, thus providing a flexible approach to the co-culture of two different types of cells. A wound could be formed automatically. Before cell seeding, different color indicators (rhodamine B, PBS solution, and blue ink) were loaded to confirm laminar flow formation inside the main microchannel. As shown in Figure 3a, three streams with clear linear boundaries indicated that laminar flow was stably linearly formed. Based on this well-controlled laminar flow, HLF and A549 cells went straightly and stably along both sides of the main channel automatically, without any mixing (Figure 3b), as expected. A blank region with clear edges gently appeared between heterotypic cells to generate a non-contact co-culture model, facilitating the observation of cell behavior in real time. It could also be used as a wound area.

Viability testing of HLF and A549 was performed on this contactless platform after 48 h. Cells maintained high viability in the microchannels after 48 h (93.6% alive for HLF and 90% alive for A549) (Figure 4), demonstrating the compatibility and feasibility of this device for further cellular assays.

With this device, different types of cells are able to be co-cultured in indirect contact for further tests such as cellular events. Passive loading facilitated rapid incubation. The size of the bare area for cell migration assay can be accurately controlled through adjusting the amounts of reagents. The edges of different compartments were neat without any cell debris or substances which would produce an unfavorable influence on cell migration.

### 3.2. Activation of HLF Indirectly Co-Cultured with A549

In order to demonstrate whether A549 can activate fibroblasts, a series of tests of cellular behaviors, such as cell morphology and cell migration, including immunofluorescence and enzyme assays were carried out on HLF. 

After HLF and A549 were successfully seeded into the co-culture device and non-contact co-cultured for 24 h, the morphological characterization of cells was assessed (Figure 3). Compared to the control groups (HLF–HLF), HLF co-cultured with A549 became nonuniform, with more protrusions stretching out extensively along the direction of A549. These protrusions, termed pseudopods, are specialized cellular structures containing an array of different proteins like matrix metalloproteinase and fibrous actins [8]. These extension structures can regulate cell membrane and cytoskeleton remodeling and are the prerequisite for the maintenance of cell motility [18]. The morphological change in HLF here was conducive to the progression and metastasis of tumor cells, implying that HLF might become activated and acquire a stronger capability for invasion in the presence of A549.

Total cell migration including cell proliferation of HLF and A549 was analyzed by tracking cells migrating from the boundary perimeter of each compartment to the middle vacant space every 4 h (Figure 5). Obviously, the migration distances of both HLF and A549 cells corresponding to cell propagation and invasion increased greatly with prolonged time under the co-culture condition compared with the other two groups. In contrast, cells moved steady in the HLF–HLF control group. However, in the HLF–A549 co-culture model, HLF started to migrate significantly at the point of 16 h, and the average migration distance of 74 μm was evidently larger than the 53.5 μm of the control (*p* < 0.05) (Figure 6a), implying that HLF located beside A549 could become activated to increase its invasion ability. Similarly, the migration of A549 was remarkably enhanced after 12 h to 62 μm at 16 h and further increased highly to 81.4 μm after 24 h when co-cultured with HLF (*p* < 0.05) (Figure 6b). This suggested that normal fibroblasts might play a vital role in tumor progression through paracrine cytokines. 

The transdifferentiation of HLF cells was further characterized after 48 h of culture. As illustrated in Figure 7, α-SMA was highly up-regulated in the HLF–A549 co-cultured group. However, only mild α-SMA expression was detected in the culture of HLF alone (HLF–HLF). This secretion distinction further suggested that tumor cells were able to activate fibroblasts into myofibroblasts.

There was no cell communication relying on direct contact between these separated cells, indicating that chemical signals coordinated the communication between HLF and A549. In order to further validate the effect of signaling pathways, the TGF-β1 content was investigated using an ELISA method. As shown in Figure 8, the results verified that TGF-β1 would be expressed by both A549 and HLF. However, the concentration of TGF-β1 secreted in the supernatant medium of HLF–A549 of 180.5 pg/mL was remarkably higher than those of the control groups: 110.5 pg/mL for A549 and 50 pg/mL for HLF (*p* < 0.05). 

Therefore, we deduced that under the condition of co-culture, HLF cells were possibly continuously influenced by high levels of TGF-β1 secreted by A549 and converted to activated fibroblasts. A549 promoted transdifferentiation of fibroblasts to acquire a favorable microenvironment for tumor growth characterized by more α-SMA. Once activated, fibroblasts characterized by a migratory spindle-shaped phenotype might generate increased TGF-β1 secretion of tumor cells with stronger invasion ability and higher expression of α-SMA. 

The transdifferentiation of fibroblasts into myofibroblasts via indirect contact might be modulated by tumor-cell-derived cytokines and further shows that tumor cells can indirectly stimulate fibroblasts and change their function to create a more survivable microenvironment. 

### 3.3. Cellular Assay on HLF Cells after Being Treated by GA and BAE

The evidence presented above elucidated that normal fibroblasts are an efficacious therapeutic target for cancer therapy. Therefore, GA and BAE were introduced to interfere with the interactions between them. 

In order to test the efficacy of GA and BAE, the total cell migration combined with the cell proliferation of HLF was noted. As shown in Fig. 9, compared with the non-drug-treated group (HLF–A549 control) and fibroblasts-only group (HLF–HLF), the fibroblasts showed a dramatic restriction in migration at different concentrations in a dose-dependent manner (*p* < 0.05) (Figure 9a). When the concentration of GA reached 20 μg/mL, the migratory ability of HLF co-cultured with A549 was controlled at a low level, as was that for HLF cultured alone. Similarly, BAE could inhibit the activation of fibroblasts (*p* < 0.05) (Figure 9b), and the best concentration of BAE for this was 10 μg/mL.

Furthermore, immunofluorescence tests were carried out on HLF cells. As shown in Figure 10, the intensity of the green fluorescence of α-SMA displayed in HLF cells treated with GA or BAE was obviously weak compared with that in the non-treated tumor-induced HLF cells, which means that GA and BAE can decrease the expression of α-SMA in fibroblasts greatly. These results suggest that both GA and BAE could suppress the effect of A549 on HLF. The effective and optimum dosages of these two drugs are 20 μg/mL and 10 μg/mL, respectively.

## 4. Conclusions

Under well-controlled laminar flow, two heterogeneous cell types interacted under non-contact within a microfluidic co-culture device. Cell co-culture and migration were examined, and protein in situ detection and cytokine detection were thoroughly performed. To elucidate the mechanism of cancer cell progression for improved cancer therapy, we accomplished in situ and label-free analysis of the interactions between tumor cells and normal fibroblasts. Our investigation revealed several valuable facts. Firstly, molecular cross-talk between tumor cells and fibroblasts was demonstrated in a contactless co-culture mode. Secondly, cytokines from tumor cells effectively transformed the co-cultured fibroblasts into myofibroblasts through indirect contact, creating a favorable microenvironment intimately associated with tumor growth and metastasis. Thus, anti-invasion cancer therapy strategies could be established through protecting fibroblasts against the influence of tumor cells. Thirdly, GA and BAE could inhibit the tumor-induced activation of fibroblasts, making them a potential source of antitumor drugs with low toxicity. Therefore, this non-contact co-culture device enables various biological assays to be performed, such as the analysis of cellular events between fibroblasts and tumor cells. Moreover, a complicated, close-to-real tumor microenvironment could be mimicked through the compartmentalization of multiple heterotypic cells within the microfluidic device for analyzing cancer invasion mechanisms and targeting anti-invasion therapeutics.

## Figures and Tables

**Figure 1 micromachines-09-00665-f001:**
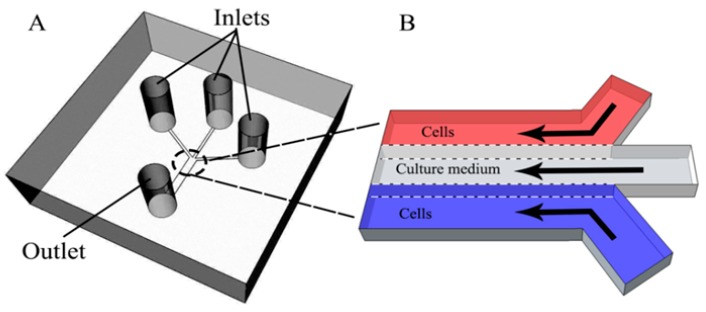
Schematic of the microfluidic platform employed for cell co-culture. (**A**) Diagram of the microfluidic device with three branching microchannels and one main microchannel. (**B**) Three parallel flows underway on the microfluidic platform.

**Figure 2 micromachines-09-00665-f002:**
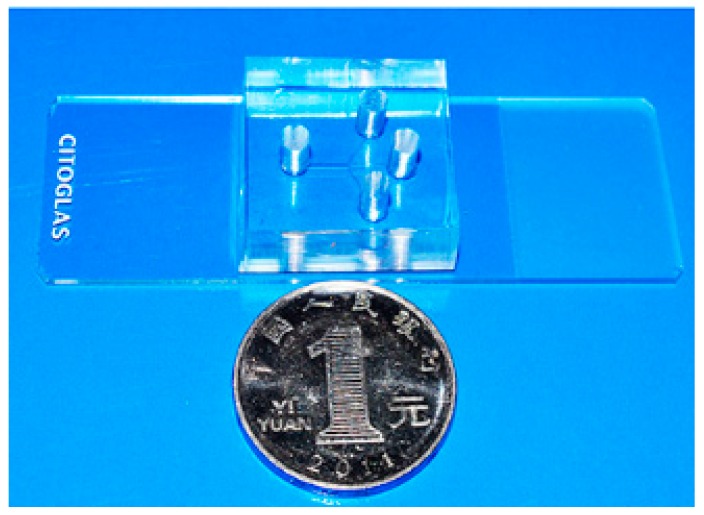
Photograph of the tiny microfluidic device.

**Figure 3 micromachines-09-00665-f003:**
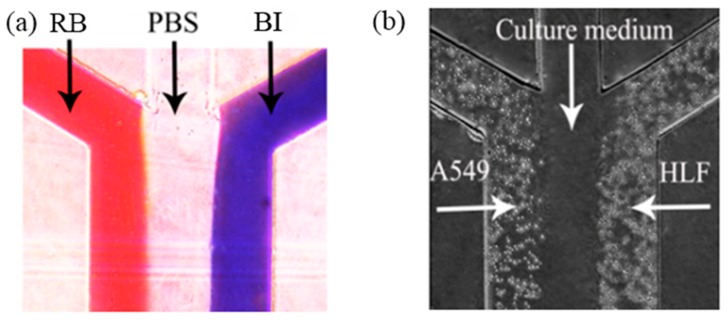
(**a**) Parallel flow testing. Red flow is rhodamine B; blue flow is blue ink; bright flow is PBS. (**b**) Different cells adhered on opposite sides of the main microchannel.

**Figure 4 micromachines-09-00665-f004:**
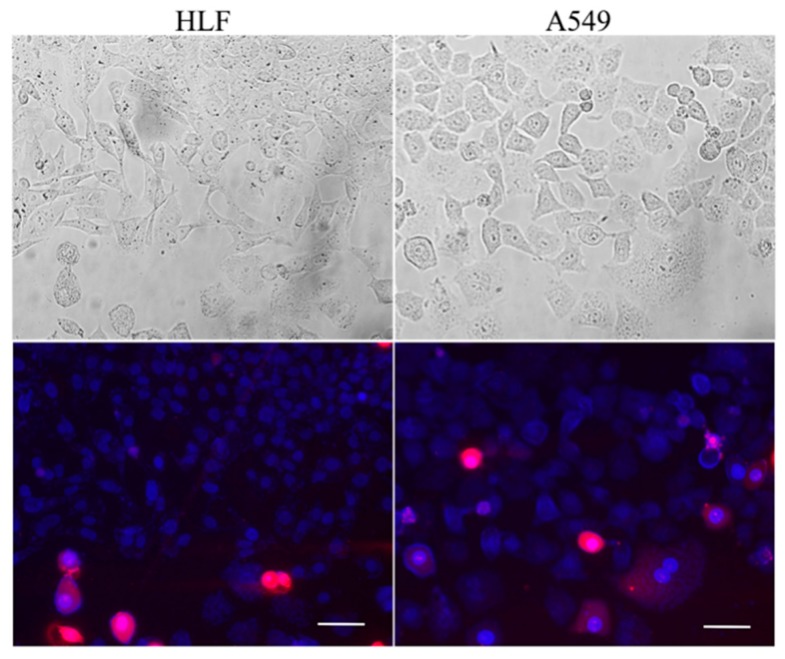
Viability of cells cultured on the co-culture device after 48 h. The top pictures are bright field and the bottom ones are merged fluorescence images. Fluorescent staining showed mostly living cells (blue) mixed with a few apoptotic cells (red). Scale bar = 50 μm.

**Figure 5 micromachines-09-00665-f005:**
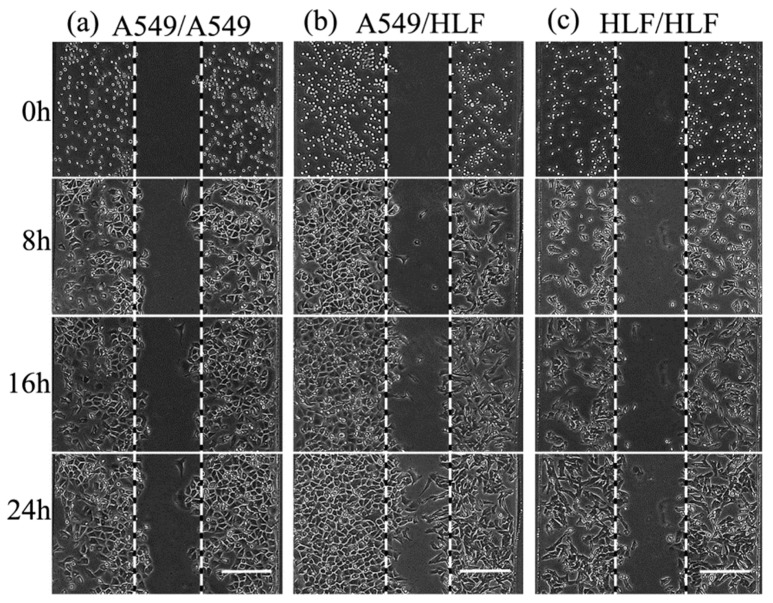
Contrast images of cells migrating on the micro-fluidic device at different times for different groups. (**a**) A549–A549. (**b**) HLF–A549. (**c**) HLF–HLF. Scale bar = 200 μm.

**Figure 6 micromachines-09-00665-f006:**
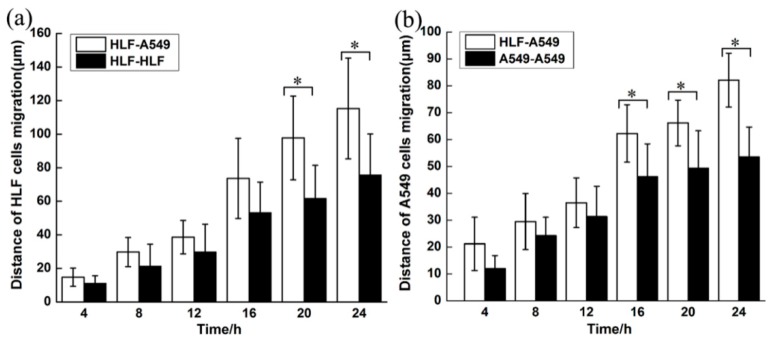
Histograms of average migration distances of cells for co-culture group compared with control groups. (**a**) Average migration distance of HLF at corresponding different times for HLF–A549 and HLF–HLF groups. (**b**) Average migration distance of A549 at corresponding different times for HLF–A549 and A549–A549 groups. (* *p* < 0.05)

**Figure 7 micromachines-09-00665-f007:**
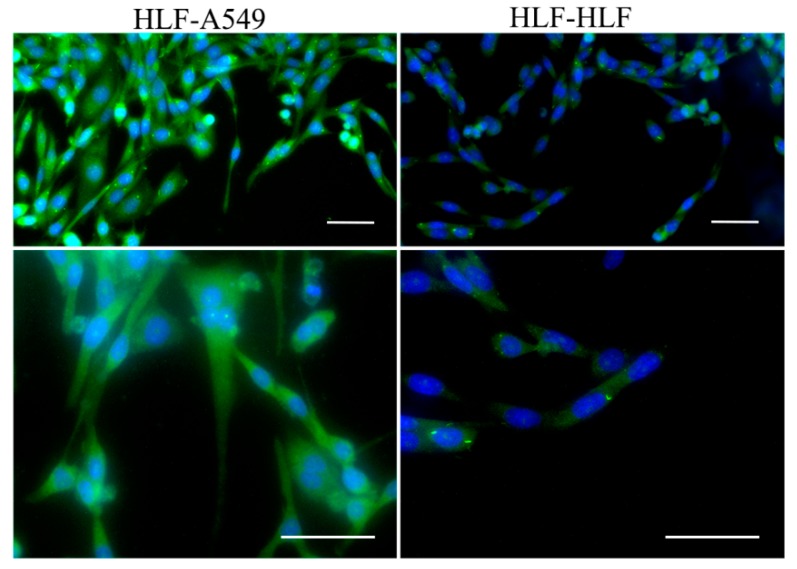
Transdifferentiation of HLF. α-SMA (green fluorescence) was used to identify activation of normal fibroblasts and the nucleus was stained blue by Hoechst 33342 (blue fluorescence). HLF could express more α-SMA after 48 h co-culture with A549 (green fluorescence). Scale bar = 50 μm.

**Figure 8 micromachines-09-00665-f008:**
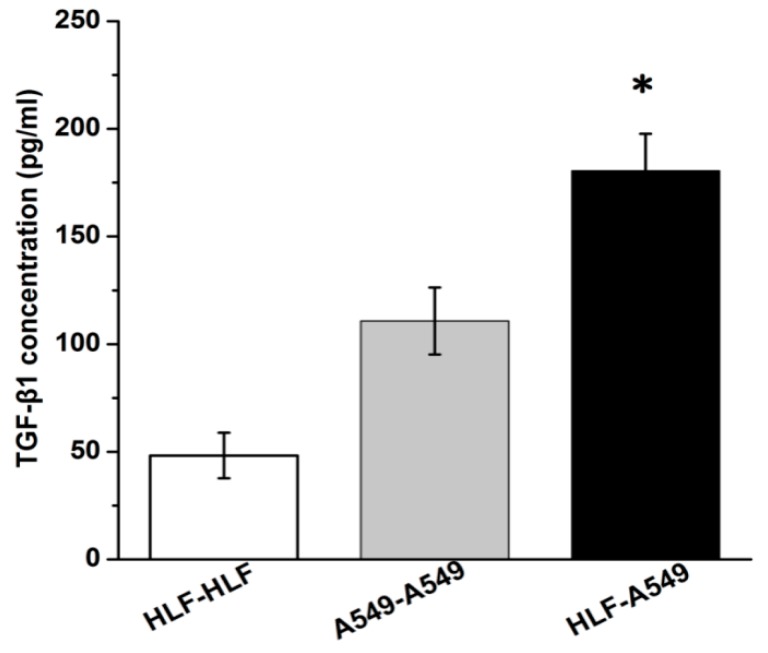
Concentrations of TGF-β1 secreted in the supernatant medium. TGF-β1 expressed by both A549 and HLF. The TGF-β1 level in HLF–A549 was remarkably higher than those in A549–A549 and HLF–HLF (* *p* < 0.05).

**Figure 9 micromachines-09-00665-f009:**
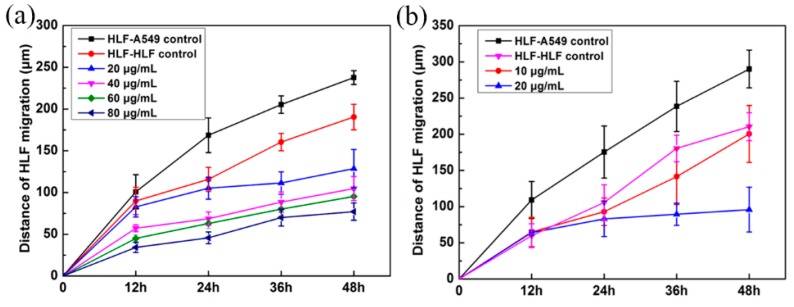
Average migration distances of tumor-induced HLF treated by gallic acid (GA) or baicalein (BAE) (* *p* < 0.05).

**Figure 10 micromachines-09-00665-f010:**
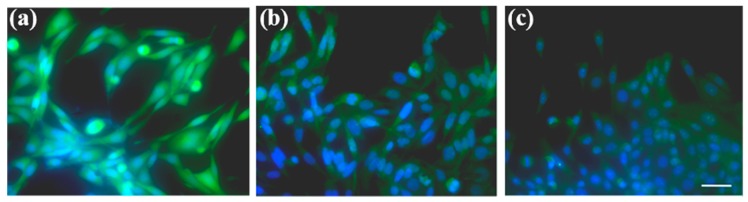
Transdifferentiation of tumor-induced HLF cells treated by GA or BAE. (**a**) Fluorescent images of HLF cells without any drugs. (**b**) Morphology of HLF cells under GA. (**c**) Cellular shapes of HLF treated with BAE.

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
