# Peer review of "In Situ Analysis of Interactions between Fibroblast and Tumor Cells for Drug Assays with Microfluidic Non-Contact Co-Culture"

_micromachines, 2018, doi:10.3390/mi9120665_

Round 1
Reviewer 1 Report
This paper introduced a microfluidic device with three converging channels to study the interaction between tumor cells and fibroblasts. Gallic acid and baicalein stimulated myofibroblasts behaviors of lung tumor cells were also studied, showing that both could inhibit activation of fibroblasts. Though non-contact coculture, the phenotypic behavior and secretion activity of tumor cells indicate that fibroblasts can be activated through paracrine signaling to create a favorable microenvironment for cancer cells. The study suggests that the non-contact microfluidic device can be useful in understanding the heterotypic cells interaction and performing anti-invasion drug essays in complex microenvironment. The paper is relatively easy to follow. However, a significant improvement in writing is needed. The reviewer also would like the following concerns to be addressed before considering further.
1. The author claimed that the microfluidic device design is novel, which is not true to the revier’s knowledge. This type of device is also called hydrodynamic focusing device, e.g., Lab Chip. 2005 Jan;5(1):14-9. 2004, J. Am. Chem. Soc., 2004, 126 (9), 2674-2675. The author should mention that.
2. Was the cell cultured in flowing media? How did the cell settle on each side of the channel without being washed away by the fluid?
3. It would be interesting to study the influence of shear force on cell migration and growth in the microfluidic device, which can significantly improve the scientific merit of the paper.
4. Line 170, more than 90% of the cells were alive after 48 hours. The author should provide details on how the viability is calculated including image analysis, what the actual percentage is. The actual percentage is more precise than “> 90% alive”.
5. The paper tends to present results in a qualitative way, e.g., line 204-206, “evidently larger than”, Line 211-214, “highly upregulated, mild expression”. Line 258: “obviously weak compared with”. The conclusion is more convincing if the results can be given quantitatively. For example, the paper can describe Fig.6 better in terms of quantitative numbers.
6. There are many writing issues, some of them are listed below. The author should spend more time working on the writing.
Line 50: detected -> experimental
Line 55, indirect -> indirect mode.
Line 85: Samples -> samples
Line 114: cells with density ->cells with a density
Line 199: redundant sign: _
Line 202-203: The meaning of “obviously, migration distances of …. “ is not clear to the reviewer.
Author Response
Reviewer #1 :
This paper introduced a microfluidic device with three converging channels to study the interaction between tumor cells and fibroblasts. Gallic acid and baicalein stimulated myofibroblasts behaviors of lung tumor cells were also studied, showing that both could inhibit activation of fibroblasts. Though non-contactcoculture, the phenotypic behavior and secrection activity of tumor cells indicate that fibroblasts can be activated through paracrine signaling to create a favorable microenvironment for cancer cell. The study suggests that the non-contact microfluiidic device can be useful in understanding the heterotypic cells interaction and performing anti-invasion drug essays in complex microenviornment. The paper is relatively easy to follow. However, a significant improvement in writing is needed. The reviewer also would like the following concerns to be addressed before considering further.
1. The author claimed that the microfluidic device is novel, which is not true to the revier’s knowledge. This type of device is also called hydrodynamic focusing device, e.g., Lab Chip. 2005 Jan; 5(1): 14-9. 2004. J. Am. Chem. Soc., 2004, 126(9),2674-2675. The author should mention that.
Response: Thank the reviewer’s kind suggestion. This type of hydrodynamic focusing device has been mentioned.
Similar design has been presented previously (Tourovskaia et al. 2005; Jahn et al. 2004 ).
Tourovskaia, A.; Fiqueroa-Masot, X.; Folch A. Differentiation-on-a-chip: a microfluidic platform for long-term cell culture studies. Lab Chip 2005, 5(1), 14-9.
Jahn, A.; Vreeland, W. N.; Gaitan, M.; Locascio L. E. Controlled vesicle self-assembly in microfluidic chanelswith hydrodynamic focusing. Journal of the American Chemical Society 2004, 126(9), 2674-2675.
2. Was the cell cultured in flowing media? How did the cell settle on each side of the channel without being washed away by the fluid?
Response: We accept the reviewer’s suggestion and thank the reviewer for pointing out. The cells was cultured in flowing media after settled down. The cells was suspended in culture media, and 3μl HLF and 3μl A549 was dropped into inlets on both outer sides. Due to gravity force, HLF and A549 go staighly laying along both sides of the microchannel. After a while when both cells are settled down, 6μl cell culture medium without cells was loaded from middle. Thus,the cell settle on each side of the channel without being washed away by the fluid
In Materials and methods, 3μl HLF and 3μl A549 were separately and simultaneously injected into inlets on both outer sides, while 6μl cell culture medium without cells was loaded from middle.
3. It would be interesting to study the influence of shear force on cell migration and growth in the microfluidic device, which can significantly improve the scientific merit of the paper.
Response: Thank you for the valuable suggestion and we accept the suggestion. The scientific merit of the paper would be greatly improved if the influence of shear force on cell migration and growth in the microfluidic device could be studied. We would do this part when suitable person and suitable software such as COMSOL are available to do this simulation, to study the influence of shear force on cell migration and growth.
4. Line 170, more than 90% of the cells were alive after 48 hours. The author should provide details on how the viability is calculated including image anlysis, what the actual percentage is. The actual percentage is more precise than “>90% alive”.
Response: Thank the reviewer for this detailed critical suggestion. The viability has been counted and described with a percentage.
Viability test of HLF and A549 were performed on this contactless platform after 48 h. Cells maintain high viability in the microchannels after 48 h (93.6% alive for HLF and 90% alive for A549) (Fig. 4), demonstrating compatibility and feasibility of this device for further cellular assays.
5. The paper tends to present results in a qualitative way, e.g., line 204-206, “evidently larger than”, Line 211-214, “ highly upregulated, mild expression”. Line 258: “obviously weak compared with”. The conclusion is more convicing if the results can be given quantitatively. For example, the paper can describe Fig.6 better in terms of quantitative numbers.
Response: Thank you for the valuable suggestion. Fig. 6 has been described in terms of quantitative numbers.
However, in the HLF–A549 co-culture model, HLF started to migrate significantly at the point of 16 h and average migration distance of 74 μm was evidently larger than that of control of 53.5 μm (p<0.05) (Fig. 6a ), implying that HLF located beside A549 could get activated to increase invasion ability. Similarly, migration of A549 was remarkably enhanced after 12 h of 62 μm at 16 h and further increased highly of 81.4 μm after 24 h when co-cultured with HLF (p<0.05) (Fig. 6b).
However, concentration of TGF-β1 secreted in the supernatant medium of HLF-A549 of 180.5 pg/ml was remarkably higher than that of control groups of 110.5 pg/ml for A549 and 50 pg/ml for HLF (p<0.05).
6. There are many writing issues, some of them are listed below. The author should spend more time working on the writing.
Line 50: detected-> experimental
Line 55: indirect-> indirect mode
Line 85: Samples-> samples
Line 114: cells with density -> cells with a density
Line 199: redundant sign: _
Line 202-203: The meaning of “ obviously, migration distances of ...” is not clear to the reviewer.
Response: Thank you for the detailed suggestion. Those writing errors have been corrected.
But physical removal of cell monolayer could damage cells, thus affect experimental result of cell migration.
To date, many researches have been performed on direct mode, but rare focused on indirect mode.
As shown in Fig. 1b, samples contained different kinds of cells were placed into three inlets.
Cells with a density of 5×106 cells/mL were loaded into the device and driven into disjunctive side areas of the main microfluidic channel.
Morphological change of HLF here was conductive to progression and metastasis of tumor cells, implying that HLF might get activated and acquire stronger capability of invasion in the presence of A549.
Obviously, migration distances of both HLF and A549 cells corresponding to cells propagation and invasion increased greatly with prolonged under the co-culture condition comparing with the other two groups.

Reviewer 2 Report
The authors report a microfluidic co-culture device to study the interaction between tumor cells and fibroblasts. The migratory behavior and secretion activity of tumor cells were quantified with HLF and A549 cells co-cultured. The effects of GA and BAE were tested using the co-culture system. Overall, the study is well-designed, and the data presentation is acceptable. The main issue is the novelty of the microfluidic system. The authors have to provide evidence that the reported system is better than existing methods, i.e., wound healing assays. Specific issues: 1. Fig.4, the percentage of apoptotic cells has to be quantified. Does the co-culture affect the proliferation and/or apoptosis of the cells? 2. Fig. 5(b), it seems after 24 hours, the two types of cells have migrated into each other, when measuring the migration, how did the authors distinguish cell types? 3. The expression of alpha-SMA needs to be better quantified, instead of, just a staining image. 4. Fig. 10, the morphology/shape change is not really visible.Author Response
The authors report a microfluidic co-culture device to study the interaction between tumor cells and fibroblasts. The migration behavior and secretion activity of tumor cells were quantified with HLF and A549 cells co-cultured. The effects of GA and BAE were tested using the co-culture system. Overall, the study is well-designed, and the data presentation is acceptable. The main issue is the novelty of the microfluidic system. The authors have to provide evidence that the reported system is better than existing methods, i.e. wound healing assays. Specific issues: 1. Fig. 4, the percentage of apoptotic cells has to be quantified. Does the co-culture affect the proliferation and/or apoptosis of the cells? 2. Fig.5(b). it seems that after 24 hours, the two types of cells have migrated into each other, when measuring the migration,how did the authors distinguish cell types? 3. The expression of alpha-SMA needs to be better quantified, instead of, just a staining image. 4. Fig.10, the morphology/shape change is not really visible
Response: Thank you for positive comments.
1) Thank you for the valuable suggestion. The percentage of apoptotic cells has been quantified,which is 6.4% for HLF and 10% for A549. The co-culture does not affect the proliferation or apoptosis of the cells.
Viability test of HLF and A549 were performed on this contactless platform after 48 h. Cells maintain high viability in the microchannels after 48 h (93.6% alive for HLF and 90% alive for A549) (Fig. 4), demonstrating compatibility and feasibility of this device for further cellular assays.
2) Response: Thank the reviewer for this question. After 24 h, the two type start to get closer, however, they have not contacted with each other. Therefore, it is not difficult to distinguish them.
3) Response: Thank you for this suggestion. We accept the reviewer’s suggestion. The expression of alpha-SMA needs to be better quantified, instead of, just a staining image. From this staining images of comparison, we should be able to tell tumor cells were able to activate fibroblasts into myfibroblasts. We would transfer the expression of alpha-SMA into quantified if a good and suitable way could be figured out.
4) Response: Thank you for this suggestion. We accept the reviewer’s suggestion. Fig.10 has been adjusted to see the morphology/shape change.

Reviewer 3 Report
Chen et al. have described work on developing a microfluidic non-contact co-culture device to analyze interactions between tumour cells and fibroblasts. They also investigate the behaviors of lung cancer cells as responses to different types of compounds that inhibit the activation of fibroblasts. Overall, this reviewer is very excited for this work since it can be applied to many other types of cancer studies and other possible applications. The authors presented the device fabrication and operation and an application for co-culturing on the device and therefore it is suitable for publication. Perhaps a review of the manuscript for spelling and grammatical errors would be sufficient before publication.
Author Response
Response: We thank the reviewer for the positive comments on this manuscript. The spelling and grammatical errors have been checked and corrected.
Round 2
Reviewer 1 Report
The author has addressed all my comments. I recommend for acceptance for publication.
Reviewer 2 Report
No further questions or suggestions.